# Implementing Pre-Therapeutic *UGT1A1* Genotyping in Clinical Practice: A Real-Life Study

**DOI:** 10.3390/jpm12020204

**Published:** 2022-02-02

**Authors:** Nicola Personeni, Laura Giordano, Angelica Michelini, Antonio D’Alessio, Antonella Cammarota, Silvia Bozzarelli, Tiziana Pressiani, Maria Giuseppina Prete, Maria Teresa Sandri, Sabine Stioui, Luca Germagnoli, Armando Santoro, Lorenza Rimassa, Rossana Mineri

**Affiliations:** 1Department of Biomedical Sciences, Humanitas University, Rita Levi Montalcini 4, Pieve Emanuele, 20072 Milan, Italy; nicola.personeni@hunimed.eu (N.P.); angelica.michelini@cancercenter.humanitas.it (A.M.); antonio.dalessio@cancercenter.humanitas.it (A.D.); antonella.cammarota@cancercenter.humanitas.it (A.C.); armando.santoro@hunimed.eu (A.S.); 2Medical Oncology and Hematology Unit, IRCCS Humanitas Research Hospital, Via Manzoni 56, Rozzano, 20089 Milan, Italy; silvia.bozzarelli@cancercenter.humanitas.it (S.B.); tiziana.pressiani@cancercenter.humanitas.it (T.P.); maria.prete@cancercenter.humanitas.it (M.G.P.); 3Biostatistic Unit, IRCCS Humanitas Research Hospital, Via Manzoni 56, Rozzano, 20089 Milan, Italy; laura.giordano@humanitas.it; 4Department of Surgery & Cancer, Imperial College London, Hammersmith Hospital, London W12 0HS, UK; 5Medical Genetics Section, Laboratory Medicine, IRCCS Humanitas Research Hospital, Via Manzoni 56, Rozzano, 20089 Milan, Italy; maria.sandri@bianalisi.it (M.T.S.); Sabine.Stioui@cdi.it (S.S.); luca.germagnoli@humanitas.it (L.G.); 6Bianalisi Laboratory, Via Mattavelli 3, 20841 Carate Brianza, Italy; 7CDI—Genetic and Cytogenetic Laboratory, Via Saint Bon 20, 20147 Milan, Italy

**Keywords:** irinotecan, UGT1A1, pharmacogenomics, metastatic cancer

## Abstract

Current guidelines recommend pre-therapeutic *UGT1A1* genotyping to guide irinotecan dosing, but the usefulness of this approach remains to be clarified. In 247 patients with advanced gastrointestinal cancers undergoing irinotecan-based chemotherapy, we prospectively performed *UGT1A1*28* genotyping and we analyzed the incidence of severe neutropenia according to genotype-guided dose reductions. Overall, 28 (11.3%) and 92 (37.2%) patients were homozygous or heterozygous *UGT1A1*28* carriers, respectively. Grade ≥ 3 neutropenia was reported in 39% of homozygous patients receiving an upfront dose reduction of irinotecan (median 40%, range 22–58%), in 20% of heterozygous or wild-type patients receiving full dose (OR_vs*28/*28 genotype_ = 0.38; 95% CI: 0.14–1.03; *p* = 0.058), and in 15.3% of those receiving a reduced dose for clinical reasons (OR _vs*28/*28 genotype_ = 0.28, 95% IC: 0.12–0.67; *p* = 0.004). Occurrence of severe neutropenia was inversely associated with dose reduction in *UGT1A1*28* homozygous carriers (OR_x10 unit_ = 0.62, 95% CI: 0.27–1.40, *p* = 0.249) and *UGT1A1* heterozygous or wild-type patients (OR_x10 unit_ = 0.87, 95% CI: 0.59–1.28, *p* = 0.478). Incidence of severe neutropenia was related to irinotecan doses and *UGT1A1* polymorphisms. Upfront irinotecan dose reductions do not reduce the burden of grade ≥ 3 neutropenia in *UGT1A1*28* homozygous carriers.

## 1. Introduction

Fluoropyrimidines (FP), irinotecan, and oxaliplatin are a mainstay in the systemic treatment of metastatic colorectal cancer [1] and other gastrointestinal malignancies including gastric, pancreatic, and biliary cancers. From 35 to 50% of patients receiving combination regimens with FP and irinotecan experience unpredictable and sometimes clinically relevant treatment-related toxicities [2]. In past years, interindividual differences in the occurrence and/or seriousness of FP- and irinotecan-related toxicities were ascribed to clinical factors, including age, sex, and performance status [3]. Irinotecan is a mildly active prodrug converted into the topoisomerase I inhibitory metabolite SN-38, eventually inactivated by the uridine diphosphate-glycosyltransferase 1 (UGT1A1) enzyme. Polymorphisms of the *UGT1A1* gene occur within its promoter sequence (TATA box), and may lead to a deficient irinotecan metabolism, resulting in drug accumulation and increased risks of toxicities. In particular, the *UGT1A1*28* allele is associated with a decreased gene expression and lower glucuronidation efficacy of the irinotecan metabolites. Associations between irinotecan toxicity and the *UGT1A1*28/*28* genotype have been reported in cancer patients [4,5,6], suggesting a significantly higher risk of severe neutropenia [7]. Given these premises, a pre-therapeutic screening for the homozygous carriers of the *UGT1A1*28* allele before the administration of irinotecan has been recommended to reduce the risk of disproportionate and sometimes fatal toxicity [8]. In addition, retrospective studies investigating the possibility to predict irinotecan toxicity with *UGT1A1* genotyping have suggested that alternative *UGT1A1* alleles/haplotypes could be more predictive of hematologic toxicity than *UGT1A1*28* [9,10]. Additionally, more recently in patients undergoing *UGT1A1* genotyping, sex, age, irinotecan dose, and treatment schedule were identified as relevant predictors of toxicity, thereby limiting the scope of *UGT1A1* polymorphisms screening [11,12,13,14]. Guidelines published by the Dutch Pharmacogenomics Working Group recommend dose reductions by 30% in patients with a known *UGT1A1*28* homozygous genotype, who are candidates to irinotecan therapy at doses of >150 mg/m^2^ [15]. Similar recommendations are found in the Italian [16] and French guidelines [17]. While most of the evidence gathered was generated on the basis of retrospective genotyping, it is still unclear whether upfront irinotecan dose reduction may mitigate irinotecan-related toxicities in individuals carrying the *28/*28 genotype. As a result, *UGT1A1* genotyping has not been routinely integrated into clinical practice yet. To this end, we undertook an analysis on the safety of irinotecan dose reductions following *UGT1A1* genotyping. In particular, here we explored the incidence of severe neutropenia in patients with gastrointestinal malignancies being treated in a palliative setting with irinotecan-containing chemotherapy regimens.

## 2. Materials and Methods

### 2.1. Patient Population

In this prospective study, we included consecutive patients who were treated at IRCCS Humanitas Research Hospital, between 2012 and 2019. The list of patients was extracted from an in-house web-based application for the management of the hospital medication life cycle, from its prescription, preparation, and its administration. Eligibility criteria included a histologically confirmed diagnosis of advanced or metastatic gastrointestinal malignancy; administration of at least one dose of irinotecan either as a single agent or in combination with other cytotoxic and/or biologic agents; and availability of clinical information and laboratory data. As of March 2016, *UGT1A1* genotyping became a standard practice in our Institution, following the publication of national AIOM-SIF recommendations [16], which recommend the pharmacogenetic analysis of *UGT1A1* before starting irinotecan therapy. This indication principally concerns patients whose clinical characteristics may lead to an increased risk–benefit ratio from irinotecan-based chemotherapy regimens. All patients carrying the homozygous *UGT1A1*28* allele polymorphism underwent irinotecan dose reduction in accordance with the guidelines [16]. On the other hand, patients with *UGT1A1*1/*1* (wild-type) and **1/*28* genotypes might receive full irinotecan doses, or reduced doses based on clinical grounds. Besides *UGT1A1*, concomitant dihydropyrimidine dehydrogenase (*DPYD)* genotyping was performed in all FP-naïve patients who were candidates to irinotecan in combination with FP. Specifically, the following *DPYD* polymorphisms were investigated: c.1905+1G>A, c.2846A>T, and c.1679T>G. Neutropenia was graded according to the National Cancer Institute Common Toxicity Criteria for Adverse Events, Version 4.3 (NCI-CTC 4.3 criteria, http://ctep.cancer.gov/reporting/ctc.html (accessed on 29 November 2021)). This study received Institutional Review Board approval (IRB No ONC-OSS-03-2020), and all patients provided written informed consent for genotyping.

### 2.2. DNA Extraction and Genotyping of UGT1A1 and DPYD

Three mL of blood were taken from all patients, stored in EDTA, and DNA was extracted from 200 μL using the Genomic DNA isolation kit (Roche Life Science, Penzberg, Germany) on a MagNA Pure LC 2.0 Instrument (Roche Life Science, Penzberg, Germany). The *DPYD* and *UGT1A1* variants were analyzed through pyrosequencing. Assays of samples for candidate gene polymorphisms were performed using the CE-IVD “Irinotecan response^®^” and “Fluoropyrimidines response^®^” kit (Diatech Pharmacogenetics^®^, Jesi, Italy), according to the manufacturer’s instructions. DNA was first amplified through RT-PCR on a RotorGene Q (Qiagen^®^, Valencia, CA, USA), and reaction products were then run on a PyroMark Q96 ID (Qiagen^®^, Valencia, CA, USA).

### 2.3. Statistical Analysis

The objective of this study was to describe the association between *UGT1A1* genotyping and grade ≥ 3 neutropenia or febrile neutropenia. Specifically, we compared the incidence of severe neutropenia in patients receiving irinotecan with and without dose reduction. Data were summarized as frequencies and proportions or as medians and ranges, and differences were estimated by the chi-square test (or the Fisher exact test when appropriate) and the *t*-test (the Wilcoxon–Mann–Whitney U test when appropriate). A logistic model was used to estimate the odds ratio (OR) and their corresponding 95% confidence intervals (CI) in univariable and multivariable analyses. The *p* value was reported considering all evaluations as explorative in nature. All analyses were performed using SAS version 9.4 (SAS Institute Inc., Cary, NC, USA).

## 3. Results

### 3.1. Baseline Patient Characteristics

Overall, 247 patients underwent UGT1A1 genotyping, with allele frequencies summarized in Table 1. 

Additionally, 179 underwent concomitant DPYD genotyping (c.1905+1G>A, c.2846A>T and c.1679T>G) when a FP-based treatment was planned in FP-naive patients (Table 2). 

This latter assessment allowed us to safely treat two patients with reduced FP doses in accordance to national guidelines [16]. All patients who were homozygous carriers of the *UGT1A1*28* allele (*n* = 28) received an upfront median irinotecan dose reduction equal to 40% (range 22–58%). On the other hand, among 219 patients who were heterozygous carriers of either the *UGT1A1*28* allele or the wild-type allele, only 56 were deemed clinically fit to receive full irinotecan doses. Their clinical characteristics are summarized in Table 3. Significant differences between the two subgroups of patients (*28/*28 vs. *1/*1 and *1/*28) were observed in terms of primary tumor location (*p* = 0.016) and baseline bilirubin levels (*p* < 0.001). 

The remaining 163 heterozygous or wild-type patients who were not deemed clinically fit for full irinotecan doses received a median dose reduction equal to 29% (range 12–55%), which was justified on the basis of a poor performance status.

### 3.2. Factors Associated with Grade ≥ 3 Neutropenia in the Whole Cohort

Grade ≥ 3 neutropenia was detected in 47/247 patients (19.0%), including five patients developing febrile neutropenia. In univariate analyses the risk of grade ≥ 3 neutropenia was higher in female patients as compared to male patients (OR = 2.94, 95% CI: 1.96–3.85; *p* = 0.030). Other clinical parameters, including age, body surface area, performance status, and *UGT1A1* genotyping were not significantly associated with the development of severe neutropenia. 

### 3.3. Grade ≥ 3 Neutropenia Associated with *28/*28, *1/*1, and *1/*28 Genotypes

Eleven patients (39.2%) carrying the homozygous *UGT1A1*28* allele developed grade ≥ 3 neutropenia despite upfront reduced irinotecan doses, as compared to 19.6% of heterozygous or wild-type patients treated with full doses (OR_vs*28/*28 genotype_ = 0.38; 95% CI: 0.14–1.03; *p* = 0.058). Of note, in *28*28 carriers, the risk of developing grade ≥ 3 neutropenia was inversely associated on a continuous scale to the irinotecan dose reduction that had been administered (OR_x10 unit_ = 0.62, 95% CI: 0.27–1.40, *p* = 0.249).

### 3.4. Grade ≥ 3 Neutropenia Associated with *1/*28 or *1/*1 Genotypes in Patients Receiving Reduced Doses of Irinotecan

Among 163 patients carrying the *UGT1A1*1*28* or *UGT1A1*1*1* genotypes treated with upfront reduced doses of irinotecan following clinical judgment, 25 (15.3%) developed grade ≥ 3 neutropenia. Compared with *UGT1A1*28* homozygous patients, the rate of grade ≥ 3 neutropenia was significantly lower (OR _vs*28/*28 genotype_ = 0.28, 95% CI: 0.12–0.67; *p* = 0.004). Again, the risk of developing grade ≥ 3 neutropenia was inversely associated on a continuous scale to the upfront irinotecan dose reduction (OR_x10 unit_ = 0.87, 95% CI: 0.59–1.28, *p* = 0.478). 

## 4. Discussion

The UGT1A1 enzyme activity is reduced in patients carrying the homozygous mutation in the *UGT1A1* gene (*28/*28 alleles); dose reductions have therefore been recommended to prevent severe adverse events associated with irinotecan [14]. Given these premises, this study evaluated the incidence of neutropenia following pre-therapeutic irinotecan dose modifications driven by the *UGT1A1* genotype. The most relevant finding is that the *UGT1A1*28*28* genotype is associated with a substantial risk of grade ≥ 3 neutropenia despite an upfront irinotecan dose reduction. Although a precise dose reduction in this patient population has not been established, our practice is consistent with current guidelines suggesting the reduction of irinotecan dose in patients with the *UGT1A1*28*28* genotype [15,16,17].

In addition, the median dose reduction we applied was in line with those indicated by previous studies that prospectively investigated dose reductions by nearly 30% when irinotecan was to be administered to homozygous carriers at doses comprised between 150 mg/m^2^ [18,19] and 180 mg/m^2^ [20]. Nevertheless, most of those studies investigated only a small number of patients with *UGT1A1*28* homozygous polymorphisms, ranging between one and seven [18,19]. As such, these numbers do not allow any conclusions. On the other hand, with a pre-therapeutic median irinotecan dose reduction equal to 40%, we observed among 27 *UGT1A1*28* homozygous carriers a 39% rate of grade ≥ 3 neutropenia. These findings are similar to those reported by Iwasa et al. who detected a 46% rate of grade ≥ 3 neutropenia in a population of East-Asian patients with homozygous genotypes (or heterozygous *UGT1A1*28* and **6* haplotypes) who received a reduced starting dose of irinotecan equal to 150 mg/m^2^ [20]. The breakdown according to *28/*28 and the *6/*6 genotypes was not provided, although homozygosity for *UGT1A1*6* has been exclusively observed in Asian populations, where the frequency of *UGT1A1*6* is higher than *UGT1A1*28* [21].

Previous studies revealed that in patients with a *UGT1A1*28/*28* genotype, irinotecan doses ≤150 mg⁄m^2^ carried a risk of hematologic toxicity not statistically different from the risk reported in patients with one or two wild-type alleles [22]. In contrast, a meta-analysis [23] reported an increased risk of neutropenia not only at medium or high doses of irinotecan, but also at lower doses, in patients with a *UGT1A1*28/*28* homozygous genotype.

The limited number of *UGT1A1*28/*28* carriers in the current study does not allow to draw firm conclusions on irinotecan dose thresholds and risk of severe neutropenia. However, we showed that this risk is inversely related to the upfront irinotecan dose reduction. The discrepancies with the aforementioned data [22] might be due to a certain degree of clinical heterogeneity (such as cancer type, therapeutic line, combination regimens). 

In the present study, the risk of grade ≥ 3 neutropenia in *28 homozygous carriers is increased when compared to patients with heterozygous (39%) or wild-type genotypes that received full dose irinotecan (20%), or reduced doses on the basis of clinical judgment (15%). Our findings are in line with a prior report from a Japanese series [18] where a non-significant increase of neutropenia incidence was also observed in patients carrying the *UGT1A1*6/*28* heterozygous genotype, despite initially reduced irinotecan doses. Similar to Caucasian patients, a reduced glucuronidation is the molecular underpinning for hematological toxicities associated with this genotype. Consistently, a recent publication by Hulshof et al. again reports on higher rates of grade ≥ 3 neutropenia among *28 and *93 homozygous carriers who receive irinotecan, despite a pre-therapeutic 30% dose reduction [24]. 

Importantly, all patients who were FP-naïve were further genotyped for *DPYD* polymorphisms and this suggests that the toxicities could not be related to concomitant FP-containing treatment. 

As expected, the concentration of total bilirubin was increased in patients with homozygous polymorphisms in *UGT1A1* compared to the heterozygous/wild-type group. 

The package insert of irinotecan approved by US Food and Drug Administration indicates that patients with total bilirubin levels between 1.0 and 2.0 mg/dL have a greater likelihood of grade 3–4 neutropenia [25]. In contrast, data for bilirubin levels > 2.0 mg/dL are lacking given that clinical trials of irinotecan excluded all patients with such baseline laboratory values. In theory, evidence for *UGT1A1* genotype-guided dosing of irinotecan should be established on the basis of a randomized controlled trial on treatment outcome, such as response rate. However, such a trial is hardly feasible because of the large number of patients needed to achieve sufficient power, which would require at least 6000 patients to be prospectively screened [14]. 

This study has several limitations, as data on concomitant therapies were not routinely captured in the current series. Besides chemotherapy, a number of drugs often administered to cancer patients may reduce the glucuronidation process, resulting in an increased cellular exposure that eventually determines specific toxicity patterns. Whereas current knowledge on the therapeutic effects of *UGT* polymorphisms mainly relates to irinotecan and mycophenolate mofetil metabolism, the clinical relevance of *UGT* polymorphisms other than *UGT1A1*28* still needs to be fully explored [26]. Furthermore, additional *UGT1* variants might modulate irinotecan-induced toxicities. One of them is the rs11563250G variant allele which is linked to a significantly decreased risk of neutropenia in patients treated with irinotecan-based chemotherapy, thereby suggesting a protective effect [27].

## 5. Conclusions

Despite the dose reduction we applied, we observed a numerically higher rate of severe neutropenia in patients with homozygous polymorphisms in the *UGT1A1* gene (*28/*28) when compared to heterozygous or wild-type genotypes (*1/*1 or *1/*28 allele carriers) with and without dose reduction. Nevertheless, albeit non-significant, our data suggest that such a risk is inversely proportional to the irinotecan dose reduction that is applied. These findings also indicate that a dose reduction does not prevent the onset of severe neutropenia in patients with gastrointestinal malignancies carrying the *UGT1A1*28/*28* genotype. However, additional measures should be considered to improve the irinotecan safety in these patients, including the prophylactic use of G-CSF among those felt to yield an increased risk of febrile neutropenia, or further dose reductions, if supported by correlative pharmacokinetic data. In conclusion, our data suggest that there is a relationship between the percentage of drug reduction and neutropenia. Genetic testing is therefore indicated to identify a group of patients at a higher risk of hematological toxicity. 

## Figures and Tables

**Table 1 jpm-12-00204-t001:** *UGT1A1* allele frequencies in the whole cohort.

*UGT1A1* Allele	Number of Patients	%
*UGT1A1*1/*1* (wild-type)	127	51.42
*UGT1A1*1/*28*	92	37.25
*UGT1A1*28/*28*	28	11.34
Total	247	100.00

**Table 2 jpm-12-00204-t002:** DPYD allele frequencies in the whole cohort.

*DPYD* Allele	Number of Patients	%
*DPYD*1/*1* (wild-type)	175	97.77
Heterozygous *DPYD*2A*	2	1.12
*DPYD* c.2846A>T	1	0.56
*DPYD* c.496A>G + *DPYD*5* [1905+1G=;1627A>G]	1	0.56
Total	179	100.00

**Table 3 jpm-12-00204-t003:** Patient demographics according to initial irinotecan dose.

Clinical Characteristics	Upfront Reduced Irinotecan Dose (*28/*28), *N* = 28	Upfront Full Irinotecan Dose (*1/*1, *1/*28), *N* = 56	*p*
Median age, years (range)	66.5 (41.1–80.0)	61.7 (36.5–80.6)	0.168
Median body surface area (range)	1.76 (1.46–2.15)	1.81 (1.48–2.35)	0.479
Female *N* (%)	12 (42.9%)	20 (35.7%)	0.525
ECOG Performance Status			0.925
-0	15 (53.6%)	27 (48.2%)	
-1	12 (42.9%)	26 (46.4%)	
-2	1 (3.6%)	3 (5.4%)	
Primary tumor location *N* (%)			0.016
-colorectal cancer	13 (46.4%)	39 (69.6%)	
-bilio-pancreatic cancer	9 (32.1%)	15 (26.8%)	
-gastric cancer	6 (21.4%)	2 (3.6%)	
Median baseline total bilirubin (mg/dL) (range)	1.16 (0.3–2.0)	0.6 (0.24–1.6)	<0.001
Median baseline neutrophil count (range)	4.25 × 10^3^/mm^3^ (2.2–23.3)	4.45 × 10^3^/mm^3^ (1.9–14)	0.755
Planned median dose of irinotecan to be administered (mg/m^2^; range)	180 (60–200)	180 (80–200)	0.489
Combination therapy ^#^	23 (82.1%)	51 (91.1%)	0.234

ECOG: Eastern Cooperative Oncology Group; ^#^ Includes the following chemotherapy regimens: Folinic Acid-Fluorouracil-Irinotecan-Oxaliplatin (Folfirinox); Folinic Acid-Fluorouracil-Irinotecan (Folfiri); Folfiri plus aflibercept; Folfiri plus bevacizumab; Folfiri plus anti-Epidermal Growth Factor Receptor antibodies.

## Data Availability

The data that support the findings of this study are available on request from the corresponding authors L.R. and R.M. The data are not publicly available due to them containing information that could compromise research participant privacy/consent.

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
