# Peer review of "Implementing Pre-Therapeutic UGT1A1 Genotyping in Clinical Practice: A Real-Life Study"

_jpm, 2022, doi:10.3390/jpm12020204_

Round 1

Reviewer 1 Report

It is well known that patients carrying UGT1A1*28 or UGT1A1*6 allele variants benefit from lower doses of irinotecan, an antineoplastic medication; however, current studies are heterogeneous. In this retrospective study of seven years, the authors compared the incidence of severe neutropenia in patients receiving irinotecan with and without dose reduction.

FDA recommends a reduction in the starting dose of irinotecan for homozygous UGT1A1*28/*28 patients, who have an increased risk of neutropenia. However, the precise dose reduction varies and requires dose modifications based on individual´s tolerance to treatment (involving genetic and clkinical factors).

I have few minor comments to the authors:

-Do the patients were taking additional comedications? Any concomitant UGT1A1 drug substrate (acetaminophen (paracetamol), carvedilol, etoposide, lamotrigine or simvastatin) or any UGT1A1 inhibitors (including unsaturated fatty acids). This should be included in the patients´ clinical characteristics.

-In a recent similar work (DOI: 10.1016/j.ejca.2021.12.009), the researchers demonstrate that UGT1A1 genotype-guided dosing significantly reduces the incidence of febrile neutropenia in UGT1A1 poor metabolizers patients treated with irinotecan, resulting in a therapeutically effective systemic drug exposure, and is cost-saving. Authors should include this reference in the discussion.

-It has been demonstrated that carriers of allele G of rs11563250 variant of UGT1A1 may have a lower risk of irinotecan-induced neutropenia, and lower total plasma bilirubin levels, suggesting that this variant is associated with an enhanced capacity for glucuronidation. Authors should discuss limitations of the study, and as they stated in the introduction there must exist alternative UGT1A1 alleles/haplotypes that could be more predictive of hematologic toxicity than UGT1A1*28.

Author Response

Response to Reviewer 1 Comments

It is well known that patients carrying UGT1A1*28 or UGT1A1*6 allele variants benefit from lower doses of irinotecan, an antineoplastic medication; however, current studies are heterogeneous. In this retrospective study of seven years, the authors compared the incidence of severe neutropenia in patients receiving irinotecan with and without dose reduction.

FDA recommends a reduction in the starting dose of irinotecan for homozygous UGT1A1*28/*28 patients, who have an increased risk of neutropenia. However, the precise dose reduction varies and requires dose modifications based on individual´s tolerance to treatment (involving genetic and clkinical factors).

I have few minor comments to the authors:

-Do the patients were taking additional comedications? Any concomitant UGT1A1 drug substrate (acetaminophen (paracetamol), carvedilol, etoposide, lamotrigine or simvastatin) or any UGT1A1 inhibitors (including unsaturated fatty acids). This should be included in the patients´ clinical characteristics.

Response: We agree with the Reviewer regarding the impact of UGT1A1 polymorphisms on pharmacokinetics of several drugs besides irinotecan. In fact, exposure to co-administered drugs may inhibit glucuronidation process resulting into an increased cellular exposure to unconjugated drugs, which may lead to a higher toxicity of the primary drug and its metabolites. Whereas the largest body of data on the therapeutic effects of UGT polymorphisms mainly concerns irinotecan and mycophenolate mofetil, data on co-administered drugs were not captured in the present database. We addressed this limitation in the Discussion section and we added a novel reference (Ref. #26).

-In a recent similar work (DOI: 10.1016/j.ejca.2021.12.009), the researchers demonstrate that UGT1A1 genotype-guided dosing significantly reduces the incidence of febrile neutropenia in UGT1A1 poor metabolizers patients treated with irinotecan, resulting in a therapeutically effective systemic drug exposure, and is cost-saving. Authors should include this reference in the discussion.

Response: We thank the Reviewer for this useful information, which we implemented in the Discussion section. Also, we have added the new reference (Ref. #24), while the remaining references have been renumbered accordingly.

-It has been demonstrated that carriers of allele G of rs11563250 variant of UGT1A1 may have a lower risk of irinotecan-induced neutropenia, and lower total plasma bilirubin levels, suggesting that this variant is associated with an enhanced capacity for glucuronidation. Authors should discuss limitations of the study, and as they stated in the introduction there must exist alternative UGT1A1 alleles/haplotypes that could be more predictive of hematologic toxicity than UGT1A1*28.

Response:  We agree with the Reviewer on the existence of additional determinants to predict hematologic toxicity to irinotecan. As such, we have addressed this issue in the Discussion section, and we added a new reference (https://doi.org/10.1038/tpj.2015.12, Ref. #27)

Reviewer 2 Report

In the manuscript entitled “Implementing pre-therapeutic UGT1A1 genotyping in clinical practice: a real-life study” the authors investigate the utility of this genotyping approach to guide irinotecan dosing in clinical practice. Below are the suggestions to improve the manuscript.

  1. The authors mention that FDA indications for the Irinotecan drug do not cover bilirubin >2.0 mg/dL. Why? The authors should discuss.
  2. In the abstract, it is mentioned that UGT1A1 genotyping was performed prospectively. However, in the manuscript (2.1. Patient population), it is mentioned that this genotyping was done retrospectively. Which one is correct? This should be clarified.
  3. In the conclusions section, the authors say that along with dosage reduction, additional measures should be taken to improve irinotecan safety in patients. Specifically, what additional measures are needed? The authors should discuss.

Author Response

Response to Reviewer 2 Comments

In the manuscript entitled “Implementing pre-therapeutic UGT1A1 genotyping in clinical practice: a real-life study” the authors investigate the utility of this genotyping approach to guide irinotecan dosing in clinical practice. Below are the suggestions to improve the manuscript.

  1. The authors mention that FDA indications for the Irinotecan drug do not cover bilirubin >2.0 mg/dL. Why? The authors should discuss.

Response: We thank the Reviewer for this observation. According to FDA indications, the use of irinotecan in patients with significant hepatic impairment has not been established in clinical trials. In fact, irinotecan was not administered to patients with serum bilirubin >2.0 mg/dL. On the other hand, in clinical trials considering weekly dosage schedules, patients with modestly elevated baseline serum total bilirubin levels (1.0 to 2.0 mg/dL) had a significantly greater likelihood of experiencing first-cycle, grade 3 or 4 neutropenia than those with bilirubin levels that were less than 1.0 mg/dL (50% [19/38] versus 18% [47/226]; p<0.001).

We tackled this point further in the Discussion section, in order to provide more in-depth information to the Reader.

2. In the abstract, it is mentioned that UGT1A1 genotyping was performed prospectively. However, in the manuscript (2.1. Patient population), it is mentioned that this genotyping was done retrospectively. Which one is correct? This should be clarified.

Response: We thank the Reviewer for this observation. Under the “Patient population” subheading, we made an amendment in order to clarify that this was a prospective study. 

 3. In the conclusions section, the authors say that along with dosage reduction, additional measures should be taken to improve irinotecan safety in patients. Specifically, what additional measures are needed? The authors should discuss.

Response: We thank the Reviewer for this observation. Additional measures might include a profilactic use of G-CSF among those patients felt to be at increased risk of febrile neutropenia, or further irinotecan dose reductions, which should be supported by correlative pharmacokinetics data.

We tackled this point in the Conclusions section.
